# A Systematic Literature Review of the Impact of Climate Change on the Global Demand for Psychiatric Services

**DOI:** 10.3390/ijerph20021190

**Published:** 2023-01-09

**Authors:** Julia Feriato Corvetto, Ammir Yacoub Helou, Peter Dambach, Thomas Müller, Rainer Sauerborn

**Affiliations:** 1Heidelberg Institute of Global Health (HIGH), Heidelberg University Hospital, Heidelberg University, 69120 Heidelberg, Germany; 2Department of Anatomy, Institute of Biomedical Sciences, University of São Paulo, São Paulo 05508-900, Brazil; 3Private Clinic Meiringen, 3860 Meiringen, Switzerland; 4Translational Research Center, University Hospital of Psychiatry and Psychotherapy, University of Bern, 3000 Bern, Switzerland

**Keywords:** climate change, mental health, psychiatric services, services demand

## Abstract

Climate Change (CC) imposes important global health risks, including on mental health (MH). They are related mostly to psychological suffering caused by climate-related events and to the heat-vulnerability caused by psychiatric disorders. This growing burden may press MH services worldwide, increasing demand on public and private systems in low-, middle-, and high-income countries. According to PRISMA, two independent reviewers searched four databases for papers published before May 2022 that associated climate-related events with healthcare demand for psychiatric conditions. Of the 7432 papers retrieved, we included 105. Only 29 were carried out in low- and middle-income countries. Twelve related the admission numbers to (i) extreme events, while 93 to (ii) meteorological factors—mostly heat. Emergency visits and hospitalizations were significantly higher during hot periods for MH disorders, especially until lag 5–7. Extreme events also caused more consultations. Suicide (completed or attempted), substance misuse, schizophrenia, mood, organic and neurotic disorders, and mortality were strongly affected by CC. This high healthcare demand is evidence of the burden patients may undergo. In addition, public and private services may face a shortage of financial and human resources. Finally, the increased use of healthcare facilities, in turn, intensifies greenhouse gas emissions, representing a self-enforcing cycle for CC. Further research is needed to better clarify how extreme events affect MH services and, in addition, if services in low- and middle-income countries are more intensely demanded by CC, as compared to richer countries.

## 1. Introduction

Climate is changing drastically and unprecedentedly [1,2,3]. Recently, the eight consecutive years from 2015 until 2022 were considered the warmest years since the beginning of temperature measurement [4,5]. Concomitantly, extreme events have become more visible and deadly [6]. Still, this threatening scenario tends to be aggravated, considering that only few countries have met their Nationally Determined Contributions (NDCs) so far, and that greenhouse gas emissions (GHG) continue to grow every year [7,8].

This wide range of climate change-sensitive variables—weather components potentially worsened by climate change (CC)—imposes consequences on human health. They were extensively explored during the past decades by the intersection between environmental and health research, but in contrast to the somatic medical conditions, much less is known about the mental health (MH) impacts of CC. Based on Verner et al. (2016), the Appendix A displays the publication numbers from the main health domains and compares them to how mental health received late scientific attention, in the context of CC [9].

The available literature could already indicate how CC can interfere on MH, as well [10]. In this scenario, there are specific reasons why MH patients are in a vulnerable condition. The main contributors are the heat intolerance created by the disorders, pre-existing social and economic vulnerability, and psychotropic drugs used as treatment [10,11,12,13].

This burden may be still intensified, due to the high rates of psychiatric disorders worldwide. Since 2005, there has been a growth of approximately 15% of the prevalence of common mental diseases, and, among youth, they occupy the leading cause of health afflictions in the United States [14,15]. The incidence during lifetime can reach levels up to 50% of the population, depending on the country [16]. Finally, once it is diagnosed, the condition imposes an important disability on the patient [17].

This health disturbance can be reflected in the overload of health systems worldwide, increasing costs for both public and private sources [18,19]. This acute effect can be measured through different methods. In a country where healthcare is effectively delivered, service utilization—emergency department visits, hospitalization, scheduled outpatients—is usually considered the most accurate method for acute needs, as it has the potential sensitivity for fluctuation according to external situations [20].

Contrarily, what happens in low- and middle-income countries is that the funding for mental illness care does not reflect the real need and corresponds to a median spending of less than 1% of their health budget [19,21,22,23]. Human resources are also scarce, and, commonly, there is only one psychiatrist responsible for thousands of patients [24]. In this case, the admission numbers, despite still being the best possible measurement, are probably fewer than anticipated, when compared to an ideal scenario. In both described cases, high- or low- and middle-income countries, the psychiatric health demand will gradually continue to pressure the systems, considering that CC tends to intensify and create new climate-related events [2,10].

Therefore, from a public health perspective, it is important to differentiate and focus on conditions that are potentially long-lasting or overwhelming for the collective. In addition, the recent terms ‘climate change distress’, ‘eco-anxiety’, and ‘solastalgia’ are terms used to describe the uncertainty and fear produced by CC, and they were found by Charlson et al. (2021) [25] to be among the most prevalent outcomes measured by the studies, even though they are often considered limited conditions [26]. By using health services as the final measure, we tend to delineate these conditions and then be more objective about the long-term burden.

One similar review, which also analyses mental health services demand, was recently conducted in Australia, with specific focus on heatwaves. The authors—Mason and Colleagues, 2022 [27]—indicate that such evidence is important to increase preparedness of health systems across the country, in order to supply and balance the demand of warmer-than-average periods.

Therefore, the purpose of this review is:To investigate the demand for mental health services caused by CC worldwide—hospital admissions (HA), emergency department visits (EDV), outpatient consultations, and ambulance dispatch—due to psychiatric symptoms, disorders, or mortality.To identify which of these disorders are responsible for the consultations, in the aftermath of CC variables.To compare health services use in low- and middle-income countries to that in high-income countries and examine how these populations are undertreated for having less access to healthcare services.

To the best of our knowledge, this is the first systematic literature review to focus on the effective health system burden correlated to CC and psychiatry.

## 2. Materials and Methods

This systematic literature review was conducted in accordance with the PRISMA guidelines (Preferred Reporting Items for Systematic reviews and Meta-Analyses) [28] and registered in PROSPERO (2022, CRD42022353023) [29].

Papers were retrieved using the applied search terms on the databases PubMed, Embase, Web of Science, and PsycInfo, which were published until 10 May 2022, without date limit.

The keywords from two categories were combined in the following search terms, using Boolean operators (detailed search terms and Boolean operators can be found in Appendix A): climate change, global warming, heatwave, hot temperature, natural disaster, psychiatric disorder, mental disorder, suicide, suicide attempt, depression, mania, schizophrenia, dementia, substance abuse, bipolar disorder, psychosis, organic disorder, anxiety disorder, neurotic disorder, neurocognitive disorder, and, lastly, post-traumatic stress disorder.

These psychiatric disorders were previously related to being susceptible to CC [10,25,30]. The search words are either MeSH terms or specific disease names listed on the International Classification of Diseases—10th revision (ICD-10th) (Appendix A) [31].

The CC variables were considered to be all the possible components that can be potentially influenced or worsened by CC: meteorological factors (mostly heat) and extreme events [2]. In this context, air pollution was not included in this review, since it is considered to be one of the causes of CC, instead of a consequence of it. Still, ‘hot temperature’, ‘heat’, or ‘warmer than average’ were temperature values above a certain threshold specified by the author (Appendix A).

### 2.1. Studies Selection and Quality Assessment

In the first phase, during the title and abstract selection, the papers were kept if selected by at least one of the two independent readers (J.C. and A.H.), through the online selecting tool for systematic reviews Rayyan [32]. Then, the selected papers were full text reviewed according to the inclusion and exclusion criteria. In the full text selection, both readers agreed on its inclusion, otherwise a third independent reader was required to decide. Detailed inclusion and exclusion criteria can be found in Table 1.

Studies that did not explicitly analyze the impact of CC variables on health services demand due to psychiatric causes (symptoms, disorders, or mortality) were excluded.

Before data appraisal, the ‘quality assessment tool for Observational Cohort and Cross-Sectional Studies’ or ‘for Case-Control studies’, depending on the design of the analyzed study, of the National Institute of Health (NIH) was used by the two reviewers, also independently, to determine the final eligibility for the included papers (Appendix A) [33]. Studies were assessed as either ‘poor’, ‘fair’, or ‘good’. The ‘poor’ quality studies were excluded. In case only one of the reviewers rated the study as ‘poor’, a third independent reader was required again. Otherwise, if the two assessments were considered at least ‘fair’, the study was included. Each question from the NIH questionnaires received 1 score. In case a study achieved less than 50% of the maximum score, it was considered ‘poor’. Between 50% and 74% were placed in the ‘fair’ category, and, finally, the ‘good’ were the ones that scored 75% or more.

### 2.2. Data Appraisal

Given the two different mechanisms by which CC may affect MH, the studies were distributed in the following major groups for data appraisal: (i) extreme events and (ii) meteorological variables. Even though a heatwave is considered an extreme event, it was placed in the last group (ii), together with hot temperature, since both act through the same mechanism on MH patients. In each of the groups, the corresponding diseases’ classification followed the ICD—10th revision.

First, the subgroup called ‘MH in general’ comprises the unclassified outcomes, in which the authors represented all of the psychiatric conditions. Secondly, the specific outcomes were grouped as follows: organic disorders (F00–F09), substance misuse (F10–F19), schizophrenia (F20–F29), mood disorders (F30–F39), neurotic disorders (F40–F48), behavioral disorders (F50–F59), personality disorders (F60–F69), intellectual disabilities (F70–F79), specific developmental disorders (F80–F89), and behavioral and emotional disorders with onset usually in childhood (F90–F99) (ICD–10th detailed in Appendix A). Still, suicide attempts or completed suicides were also listed separately. The last classification was called ‘mortality’, and corresponded the cases of death linked, by the authors, to a MH cause.

In case papers addressed multiple human disorders, the only retrieved data was the one corresponding to MH.

## 3. Results

We performed an extensive literature review and could rely on a relatively large number of studies (*n* = 105). The vast majority of the studies here comprised were ‘fair’ in quality (*n* = 91), and only 1 was graded as ‘poor’ and, therefore, excluded from the analysis. Still, 14 papers were rated as ‘good’.

From the 7432 papers searched via four databases (PubMed = 1183, PsycInfo = 752, Embase = 2286, and Web of Science *n* = 3211), we included 105 for data extraction (Figure 1). The general information from included studies can be found in Table 2, including the number of each type of health service that was approached. A more detailed table is included in the Appendix A. The CC variables were found to be predominantly heat and heatwaves (*n* = 92; 87.6%). Only one study reported other meteorological exposures, snow and rain, which they reported to be associated with CC. Few articles (*n* = 12; 11.4%) studied the influences of extreme events on psychiatry.

The Figure 2 displays the number of included articles according to the year of publication. This review found, as expected and previously discussed, an increase in publication numbers, especially after 2016. Until May 2022, a total of 12 studies had already been published and are included here, following the positive trend.

Twenty-nine studies (27.5%) were carried out in low- and middle-income countries. China contributed the most, with 19 publications. Figure 3 displays how the studies are spread in the world, evidencing the central role of high-income countries.

Hospital admission (*n* = 34) and completed suicide (*n* = 34) were found to be the most studied types of services. Even though suicide is not a service per se, it certainly requires specific sectors of the health system to be addressed. Emergency department visit was, as well, often studied (*n* = 20), followed by mortality (*n* = 15). Together, phone calls, ambulance dispatch, outpatient visits, and other types of provided healthcare were poorly researched (*n* = 10) and could, in the future, receive more focus from science. In addition, no significant difference among them was found, and it seems that healthcare services are equally demanded. Figure 4 lists the measure of effect and displays them according to the used service.

### 3.1. Extreme Events (n = 12)

Twelve papers (*n* = 12; 11.4%) correlate CC-related extreme events with psychiatric consultations. The significant results are presented in the following Table 3, considering a confidence interval of 95%. They are divided by type of service used and by the MH condition. In addition, the disasters responsible for the results can be found after each confidence interval. It is important to point out that the authors often presented a great range of results, given the different lags approached. Therefore, the results displayed in Table 3 represent the highest impact found by each of them, positive or negative.

The disasters were hurricane (*n* = 4), drought or dry weather (*n* = 3), floods (*n* = 4), extreme precipitation or extreme wet weather (*n* = 2), bushfire (*n* = 1), earthquake (*n* = 1), severe winter weather or ice storm (*n* = 2), dust storm (*n* = 1), tornado (*n* = 1), and severe storms (*n* = 1).

In the aftermath of these extreme events, the most strongly affected MH conditions were suicide, mood—which includes Major Depressive Disorder—and neurotic disorders (anxiety, post-traumatic stress disorder, etc.). Therefore, these are the main responsible causes for the acute health service demand, showed by the studies. Substance misuse, schizophrenia and behavioral disorder were approached by fewer studies, but also showed to demand medical care.

The detailed results are described below (i–vi). They were divided in ICD-10th groups and placed according to the strength of evidence.

(i)Suicide behavior (*n* = 9)

Suicide behavior (attempt or completed suicide) showed the most consistent increase after CC-related extreme events among all studies. From the nine articles, only one showed no statistically significant increment on suicide [136].

Also, when compared to other MH conditions, suicide showed the highest risk ratio for EDV immediately after the disaster (1.68 *, 1.54–1.82), and the risk continued to rise during the whole period of study, until three years later [126].

One study found a relative increase of suicide behavior only during extreme wet weather (18.7% *, 6.2–31.2) and not during the dry [133]. In addition, farmers were particularly searched: one paper showed relative risk of 1.15 * (1.08–1.22) in male farmers when drought intensified [135]. In contrast to males, the risk for female farmers was significantly reduced [135]. In another study, the general population showed a RR of 4.23 * (1.28–13.93) due to moderate drought [134].

Lastly, it was reported that extreme large events increased the incidence of suicide, but in case of less damaging ones, the incidence decreased in the aftermath [131].

(ii)Mood disorders (*n* = 4)

The four papers found significant increase on mental health services demand for mood disorders. One showed a risk ratio for EDV of 1.59 * (1.39–1.80) right after the disaster, but it rapidly declined. On the other hand, HA appeared significantly higher one year later and, after three years, the risk ratio of hospitalization was still 1.67 * (1.47–1.87) [126].

Also, Major Depressive Disorder (MDD) was the only subtype analyzed. One study showed an incidence rate ratio of 2.57 * (1.60–4.14) of MDD in psychological services [126]. The two others pointed an increase in service use varying from 44% [15] to 245% [129]. MDD imposed a greater chance of medical need when compared also to anxiety disorders (see next section).

(iii)Neurotic disorders (*n* = 4)

All of the studies that analyzed anxiety disorders (*n* = 4) found a positive correlation, showing an important increase in health services demand. One found 267 and 307% increase in incidence for elderly and non-elderly, respectively [129], while another one found an incidence rate ratio of psychological services of 2.06 * (1.21–3.49) [126]. One paper showed that EDV were not significant or even reduced until three years after a disaster, but contrarily, HA increased during most of the period, and two years later the relative risk was 1.15 (1.11–1.20) [126].

Adjustment disorder was not consistent. One study did not find any increase both for EDV and HA [126]. In opposition, two others pointed 62% more Medicaid use [15] and 160–200% incidence in health centres [129]. Only one paper studied post-traumatic stress disorder (PTSD) specifically, and the incidence for non-elderly in health services was 350% higher [129].

(iv)Substance misuse (*n* = 2)

For elderly, the risk ratio for EDV due to substance misuse was the second highest, only lower than suicide behavior. Immediately after the disaster, the risk ratio was 1.44 * (1.23–1.65), reduced after three months, but increased again, maintaining high levels until three years after the extreme event (1.19 *, 1.11–1.26). Hospitalizations appeared three months 1.12 * (1.04–1.21) and were significant until one year, then, reducing again [126]. One studied that analyzed Medicaid services in the USA showed 66% increase in services use, when comparing the numbers before and after an important flood [15].

(v)Schizophrenia (*n* = 1)

Schizophrenia (HA) was correlated with extreme precipitation. The result was positive for both urban and rural areas. The relative risk increased already at lag1 (1.056 *, 1.003–1.110), peaked at lag8 (1.072 *, 1.033–1.113), and remained significant until lag17 (1.039 *, 1.004–1.075). The fraction of HA attributable to extreme precipitation was 3.42% * (2.40–4.06). Still, this number was higher for men aged less than 39 years old [127].

(vi)Behavioral disorders (*n* = 1)

Health services in general received a higher incidence of 318 and 356% for non-elderly and elderly, respectively, due to insomnia (behavioral disorder) [127].

### 3.2. Meteorological Factors (n = 93)

Temperature was by far the most approached measure (*n* = 92; 86.6%) and, therefore, it will be our focus on this section (Figure 5). The ‘warmer than average’ factor was usually reported as high percentiles in relation to the threshold (Appendix A). Heatwave (HW) was also commonly analyzed (*n* = 18; 17.1%). Different definitions of HW were used, varying in terms of intensity (temperature) and duration. Five studies opted for the ‘diurnal temperature range’, whose definition is the numerical difference between the highest and lowest temperature values in 24 h. Other temperature-related factors were temperature variability, temperature change between neighboring days, daily excess hourly heat, and irregular daily variation. Finally, one study analyzed only snow and rain—among the CC sensitive variables [81], and another one included rainfall in the analysis [84].

The most strongly affected specific MH conditions were suicide behavior; schizophrenia; substance misuse; organic, mood and neurotic disorders; and, lastly, mortality. The Figure 6 shows the proportion of significant findings, divided by disease subgroups, considering a confidence interval of 95%.

The majority of studies found significant and positive evidence that MH diseases in general increased the risk for health service need, due to hot weather. The most studied services were HA, EDV, suicide, and mortality. One study [34] found that approximately 14% of all the emergency admission numbers for mental disorders were attributable to hot temperatures. The delayed or lagged effects showed an increase, according to different studies, until 5 days (RR 1.43 *) [46], 7 days (RR 1.28 *) [51], 11 days (RR 1.84 *) [54] and 14 days (Percent change in risk of 22% *) [60]. Likewise, the same-day effect was significant: RR 1.158 * for EDV [34] and 1.229 * for HA [52].

A longer HW (7 days) imposed a higher risk (RR 1.36 *) of HA than a 3-day (RR 1.15 *) and a 1-day HW (RR 1.04 *) [38]. According to one other study [59], HW intensity seemed to cause an even higher risk—62% (RR 1.62 *)—of hospitalization.

A study in China, during heatwave periods, pointed to important subgroups that were at higher risk of MH care need: residents of urban areas (OR 1.523 *, 1.233–2.349), outdoor workers (OR 1.714 *, 1.198–2.398), and single patients (OR 1.709 *, 1.233–2.349) [47].

Mortality was approached in thirteen studies, and eleven of them presented evidence on how MH is a risk factor for dying on hot days. A 1 °C increment in temperature increased the odds of dying in the general population and in MH patients by 1.9% (OR 1.019 *) and 5.5% (OR 1.055 *), respectively [88]. HW were often explored, and the percent change in cause was 29.7% * [95]. Four studies pointed out that elderly patients were more vulnerable [34,35,38,92]. One study found that poverty enhanced the risk of mortality during hot days [86].

Finally, Table 4 displays the main results. The significant numbers can be linked to which type of service was demanded and due to in which disorder group this demand took place.

Despite being included in the analysis, disorders of adult personality (*n* = 5) were not found to be related to heat. Behavioral disorders (*n* = 6) showed no consistent result, and five out of the six studies found no relationship between those variables, while one could point a positive correlation between temperature (PET) and admission numbers [49]. Additionally, only one study found that mental retardation (*n* = 4) was affected by HW [38].

(i)Suicide behavior (*n* = 33)

Suicide was the most studied outcome, and approximately 94% of the papers (*n* = 31) found a statistically significant relationship between temperature and suicide behavior: attempted or completed. Exceptionally, two of them evidenced a reduced risk of suicide during extreme hot temperatures, with a RR of 0.97 * (0.95—0.99) [90] and average decrease (according to the authors) of IRR 0.94 * between male and females [97], while the others pointed that heat increased suicide behavior. Studies showed a large range of significance: from IRR 1.06 * (1.01–1.12) [37] to RR 1.35 * [111]. Females had a higher risk, according to one study [102].

Heat increased violent suicide attempts and completed suicide, but not the non-violent behaviors in three studies [104,116,117]. One other study showed evidence that both methods were significantly higher, but still pointed out that the temperature threshold associated with increasing violent methods was lower (30.3 °C) than the temperature needed to significantly increase non-violent suicides (32.7 °C) [103].

Ambulance dispatch was measured by one study, during different intensities of heatwave, and the RR was found to be relatively high, varying from 3.70 * (1.00–13.66) to 4.53 * (1.23–16.68) depending on the HW intensity [115]. Another ambulance dispatch-based study showed a lower, but still positive, number: RR 1.11 * (1.07–1.15) [98].

Still, one study showed the indirect effect of temperature (increment of 1 °C) on a 3.6% decline of agricultural productivity, leading to a 4.8% increase in risk of suicide among farmers [108].

(ii)Schizophrenia (*n* = 24)

Seventeen studies (65.38%) found positive and significant results, showing these patients are more vulnerable to heat and, therefore, have a higher chance of needing healthcare services. Two studies presented both same-day and cumulative risk assessments. They found a higher risk when the heat was sustained up to seven or eight days (RR 2.49 *, 1.69–3.69 and 1.37 *, 1.168–1.614), while one-day heat distress brought relatively less risk: RR 1.10 * (1.03–1.17) and 1.06 * (1.019–1.106), respectively [45,71]. To mortality, schizophrenia was not significantly related.

In addition, it seems, from another study, that the heat intensity of a HW brings more risk than its duration, with a RR of 1.50 * (1.20–1.86) and RR:1.14 * (1.01–1.30), respectively [59].

Finally, there is also evidence that large temperature variation within the day (maximum minus minimum T) may increase hospital admission up to 22% (RR 1.22 *, 1.08–1.37) [76].

(iii)Mood disorders (*n* = 23)

Sixty-two percent of the papers (*n* = 15) presented evidence of the influence of temperature on outpatient appointments (OR 1.32 *, 1.08–1.62) [36], emergency department visits (IRR 1.07 *, 1.05–1.09; RR 1.05 *, 1.01–1.09; and RR 1.33 *, 1.03–1.71) [37,39,51], and hospitalization (RR 1.34 *, 1.05–1.71) [35]. One showed that rainfall has negative effect on mania [84].

From the six studies that specifically researched bipolar disorder or mania, four of them found positive correlations with an increase in healthcare demand [80,81,82,83], while two did not present significant results [84,85].

Depression, another type of mood disorder, was not found to be at significantly higher risk from HA, according to one paper [83]. Contrarily, long-term exposure to higher temperatures (residing in areas with average temperatures above 23 °C) increased the risk by 7% of developing MDD per 1 °C of increment (hazard ratio 1.07 *, 1.02–1.12) [79].

(iv)Organic disorders (*n* = 17)

Thirteen of the studies (76.47%) that analyzed organic disorders found a positive association between temperature and medical needs or even death. Patients with organic disorders had 31% more risk of being admitted to an emergency department [51] and an odds ratio of 8.33 * of being hospitalized [68]. The risk of mortality was 3% higher [11].

Alzheimer’s disease was specifically studied and showed a significant RR of 1.30 * (1.12–1.52) [65] for hospital admission and an increase in RR of 269% * (76–665%) [66] for mortality.

(v)Substance misuse (*n* = 16)

Twelve papers (70.58%) presented statistically significant association between temperature and HA or EDV, where the RR varied from 1.13 * (1.00, 1.27) for hospitalization [35] and 1.30 * (1.18–1.42) for emergency visits [51]. For intense HW, the RR was even higher, displaying 221% * higher risk of hospitalization [59].

Phone calls requiring healthcare for substance misuse were also measured: the RR was 1.08 * (1.03–1.14) [63], and the risk of death increased 8% in patients diagnosed with alcohol misuse. In those who had used other substances (excluding alcohol), the risk increased to 20% [11].

(vi)Neurotic and anxiety disorders (*n* = 14)

Eleven out of fourteen studies (78.57%) pointed out that neurotic disorders influenced healthcare demand due to hot temperatures. HA did not appear to be influenced [35], but for EDV, the RR varied from 1.05 * (1.002–1.099) [46] to 1.27 * (1.19–1.36) [51].

The odds that patients scheduled a medical appointment after a cumulative heat exposure was 30% higher (OR 1.30 *, 1.08–1.58) even at lag 9, suggesting that not-emergency services, such as scheduled appointments, may receive a delayed demand [36].

(vii)Behavioral and emotional disorders with onset during childhood/adolescence (*n* = 4)

Two of the studies pointed a higher risk of healthcare need, with an IRR of 1.11 * (1.05–1.18) for EDV [37] and RR 1.29 * (1.09–1.54) [55] for HA. A third study, in contrast, reported an important reduction of the number of emergency room visits after a heatwave: IRR 0.578 * (0.349–0.955) [42].

(viii)Disorders of psychological development (*n* = 2)

The autism spectrum disorders were found to cause additional susceptibility to heat and, therefore, demand emergency department visits (IRR 1.64 *, 1.086–2.480) [42] and hospital admissions (RR 1.29 *, 1.09–1.54) [55], with relatively elevated risk.

## 4. Discussion

The findings from the 105 studies pointed to a high probability, according to strong evidence, that more consultations will be necessary in public and private systems. We could not find any evidence supporting that the distinct types of health services are differently affected. Still, ambulance dispatch, telephone calls, and outpatient visits were seldomly measured and need to be better addressed in the future, but were, likewise, positively associated with CC variables. Completed suicide and mortality were strongly affected. Even though these two conditions do not require common consultations at health facilities, they do demand other sources of assistance to be carried out.

We identified that elderly people—above 60 years of age—were among the most vulnerable group for both extreme events and meteorological conditions [35,38]. Poverty was found to enhance the risk of heat-related death, while male farmers were particularly at risk due to extreme events. Residents of urban areas, outdoor workers, and single patients were also reported to be at higher risk [47]. The increase in health service demand was most significant until a lag 5–7 [45,46], but some studies show the effect up to 14 [60], 21 [41] and even 30 days [45].

We also found that a great range of psychiatric disorders, mostly from F50 to F99, were not properly approached, and that the scientific evidence for them is still poor. Even though these conditions are less prevalent at healthcare facilities, future attention and research on the topic are needed.

One specific disease that received less attention than expected was post-traumatic stress disorder (PTSD). Despite being broadly discussed, only one study focused objectively on if PTSD demands healthcare after an extreme event (129). This important gap of knowledge should be addressed in the future, as well.

The lack of extreme event-related studies that addressed specifically health services use was another identified gap in the scientific evidence. Most of the studies about extreme events measured the distress over the affected population through questionnaires in the communities, instead of through health assistance needs. These methodologies were removed from this review, because even though they corroborate the important mental distress these people go through, they do not necessarily reflect long-lasting suffering and more consultations in health systems. As found by Lai et al. (2021) [26], most of the symptoms in the direct aftermath will not become chronic and burdening diseases. Therefore, it is likely they do not demand intense healthcare after all.

When comparing the consultation rates in low-, middle- and high-income countries, we did not identify any significant difference, meaning that the increase rate was similar among them. We hypothesized that, given the higher vulnerabilities present in poorer countries, the numbers found would possibly be larger if their health systems were adequately prepared to receive more patients. For a proper comparison, more studies addressing healthcare use in poorer countries are needed.

Few studies found negative associations linking CC to healthcare use [90,97,126]. Begum et al. (2022) linked Hurricane Sandy to less EDV for anxiety. Contrarily though, for HA, the same article reported a significant increase. We hypothesize that the so called ‘harvesting effect’ may be responsible for a forward shift in one category of healthcare visits, inducing a consequent reduction in the others.

This increase in health demand will likely overwhelm health systems and pressure costs of healthcare. As an example, one article using Medicaid data pointed out 8–10% more costs after a flood [19], especially as extreme events become more frequent and temperatures rise even more. According to the United Nations Office for Disaster Risk Reduction, there were 389 climate-related events in 2020, a number above the previous annual average (368). In addition, one of the latest available reports shows a probable increase in temperatures of 2.4 °C by 2100, provided that all countries implement their nationally determined contributions (NCD) [137,138].

In addition to the financial costs, this acute increase in demand can create an emergency situation in health facilities, with shortage of drugs, beds, human services, and other basic resources. In a much bigger proportion, this emergency picture happened during the coronavirus pandemic, mostly in less prepared regions.

Furthermore, the carbon footprint released from the healthcare services is, already nowadays, one of the biggest contributors of CC [139]. The data we report here corroborate that these services will be more intensively utilized, leading to a self-reinforcement cycle that promotes proportionally greater greenhouse gas emissions into the atmosphere.

Some characteristics of mental health patients—directly or externally related to the disorder—may lead to this higher healthcare demand. The responsible mechanisms for that are important to be delineated. The aim is to point out a pathway to policymakers and medical professionals where policies could effectively act, avoiding consultations for development of a new disorder, for disease exacerbation, heat-related disorders, suicide attempts, and, still, prevent mortality.

Firstly, climate-related events frequently cause mental suffering. These circumstances may lead to punctual and self-limited mental distress or to a chronic disorder [10,19]. In general, the two possible scenarios are determined by a community’s pre-existing vulnerability and by the adaptation and recovery power delivered through public policies [140].

Furthermore, heat-related vulnerability is promoted by some psychiatric disorders through inadequate heat dissipation, which facilitates a dangerous state of hyperthermia in the body [141]. Still, additional heat intolerance is caused by the use of certain psychotropic medications—antipsychotics, antiparkinsonians, hypnotics, anxiolytics, and opiates [12,13,117]. A further mechanism is related to disease exacerbation (acute phase of a disorder). It seems that neurotransmitters’ imbalance, induced by different weather variables—temperature, humidity, and luminosity—leads to the exacerbation of some disorders or to the triggering of violent and suicidal behavior [10,12,22]. As an example, an observed condition was suicide attempt, whose association with heat was reported to be of statistical significance. It seems that patients who previously attempted suicide have indeed a higher susceptibility to heat. In one study, differently from the first-attempt patients, who were vulnerable during the warm season only, the patients who had tried suicide multiple times were at higher risk during the entire year, in cases in which the temperature rose 5 °C [121].

Additionally, the cognition impairment caused by some mental disorders may also increase heat-susceptibility. There may be a lack of perception about the need to go to a cooler area or to take off extra layers of clothes to cool off the body [141].

At advanced ages, the delay and impairment to control heat maintenance is physiologically explained, since the receptor system and also the areas in the brain responsible for this function lose their efficiency over time, facilitating a higher body temperature [141].

Mentally ill patients as well as the elderly population have both a higher chance of being social isolated or living in nursing homes, which has also proved to increase risk of heat-related deaths or heat stress [11].

Similarly, poverty was found to further enhance the risk of heat-related death [86]. A possible reason for this susceptibility could be poor housing conditions. As previously discussed, low- and middle-income countries accounted for less than one-third of the studies. In addition, these regions rely on a less prepared healthcare system, which, nowadays, cannot accommodate the demand. In the future, in the absence of effective policies, these patients might become undertreated.

Finally, addressing not only the mechanisms surrounding the disorders, but also the socioeconomic vulnerabilities in the background may avoid the acute need for healthcare services.

## 5. Conclusions

This review is the first one to systematically assess and analyze the impact of CC on psychiatric health systems worldwide.

We found, based on strong evidence, that warmer temperatures, including heatwaves, can increase the demand uniformly for all types of medical care. Based on fewer number of studies, we also found that extreme events increase demand for all types of services. We support the hypothesis that, if public policies act to increase the recovery power of susceptible communities, fewer of these aftermath symptoms—anxiety, fear, and sadness—will become burdening disorders, and, therefore, lower numbers of EDV, HA, etc., will be required.

In this context, we found and listed three main gaps in the literature, which need to be properly addressed by scientific research in the future: (i) the low number of studies from low- and middle-income countries measuring health services utilization due to CC; (ii) the lack of studies carried out in the aftermath of extreme events using objective measures, as HA or EDV; and (iii) the poor approach of less common mental disorders. They are listed from F50–F99, in ICD-10th.

The lack of studies from low- and middle-income countries prevented us from measuring uniformly across the globe how health systems are affected, since the evidence is derived mostly from richer countries. Even though we found no difference of service utilization between low- and middle- when compared to high income-countries, we raise the concern that these assistance numbers from poorer countries are likely to be underrated. This could be explained by the fact that they do not have the same budget allocated to healthcare and that the population may face difficulties—financial, geographical, or transportation—in accessing health facilities.

Considering the continuous increase in numbers of psychiatric patients, individuals and health systems may witness more significant overwhelm than the one presented here, as temperatures rise and extreme events become more frequent. Therefore, additional public policies are the most effective way to help populations adapt, reducing the health impacts and, finally, avoiding the extra demand on public and private healthcare systems worldwide.

## 6. Strengths and Limitations

The use of mental health services, in terms of extra demand caused by CC, is an important strength. Still, we acknowledge that the use of services requires, firstly, the existence of functional and effective services, and these might not exist in some studied countries. Additionally, cognitive and educational access is needed for a patient’s clarity regarding when to seek healthcare. Therefore, this SLR may not reflect the magnitude of the impact of CC on MH disorders.

It has been pointed out that the lack of studies about extreme events and MH care use did not permit a vast analysis, as was the case for the ‘meteorological section’. There is a need to better elucidate this issue and the difference between the punctual distress after extreme events measured by community visits and, effectively, long-lasting and burdening disease.

Given the extensive number of studies and results we had access to, the forest plot (Figure 5) is based on significant results and on relative risk-based studies only. Despite representing the biggest part of studies, we acknowledge that it may have prevented the reader from having another perspective.

Finally, the use of the ‘NIH Quality Assessment Tool’ is considered, by some authors, as not being the most appropriate tool. We acknowledge that, by having chosen this tool, we might have incurred potential bias. Nonetheless, all efforts were made to avoid this issue.

## Figures and Tables

**Figure 1 ijerph-20-01190-f001:**
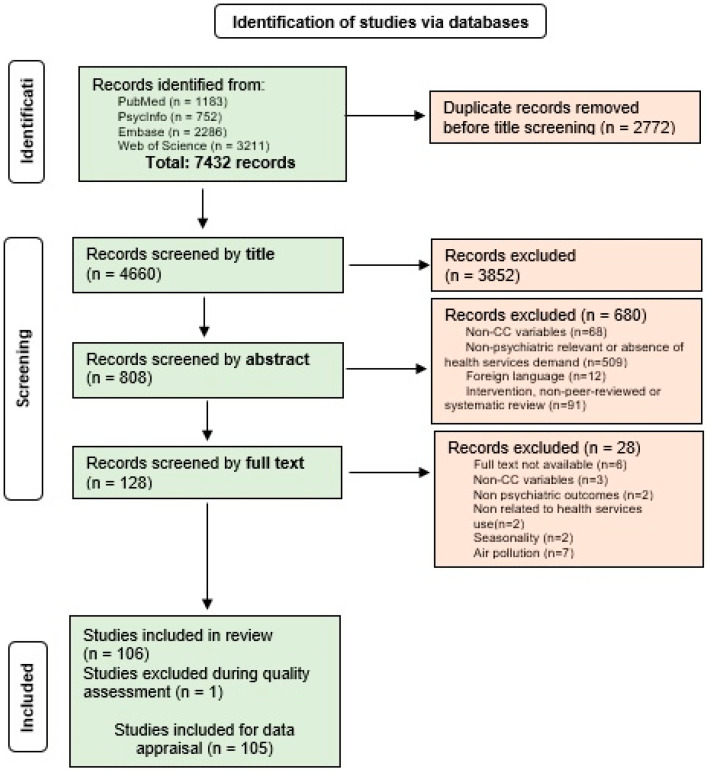
PRISMA diagram for Systematic Literature Reviews [28].

**Figure 2 ijerph-20-01190-f002:**
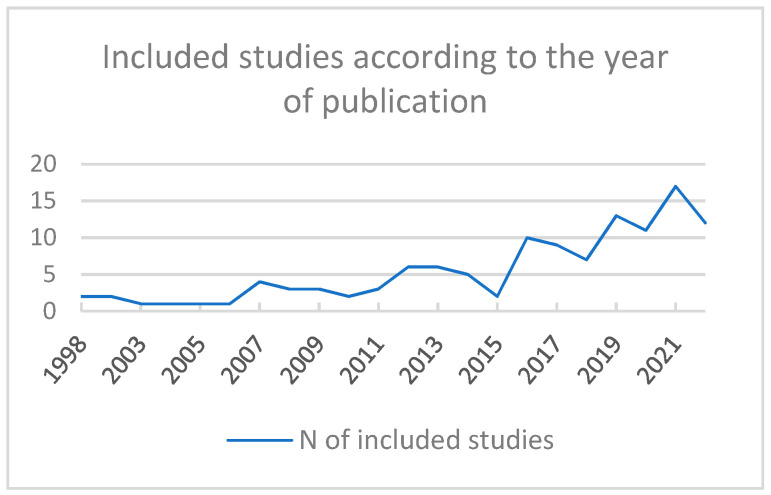
Included studies according to the year of publication.

**Figure 3 ijerph-20-01190-f003:**
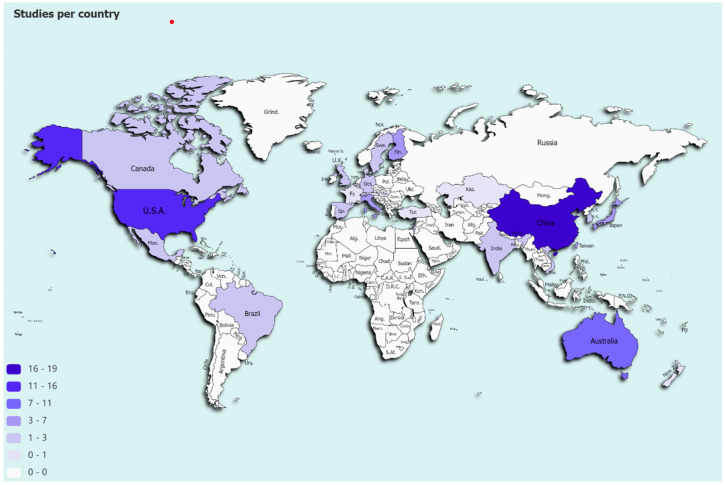
Global distribution of studies. Note: white color corresponds to missing studies in that country. Created with: paintmaps.com, accessed on 10 October 2022.

**Figure 4 ijerph-20-01190-f004:**
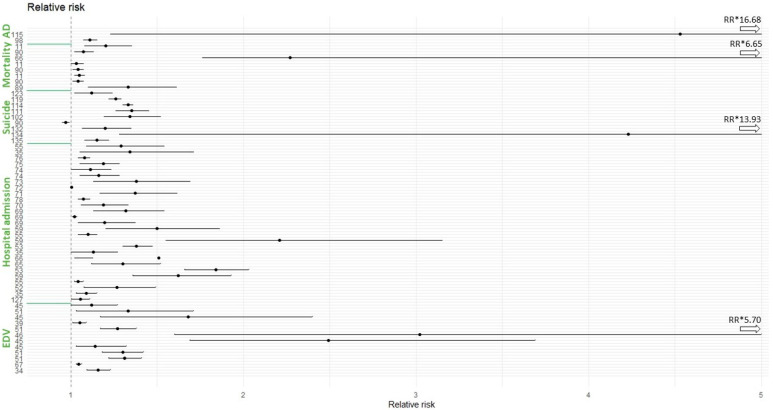
Forest plot of significant results (relative risk), divided by type of demanded service. The numbers on the *y*-axis represent the study references. Abbreviations: AD: ambulance dispatch; EDV: emergency department visit. RR * represents the respective RR upper limits.

**Figure 5 ijerph-20-01190-f005:**
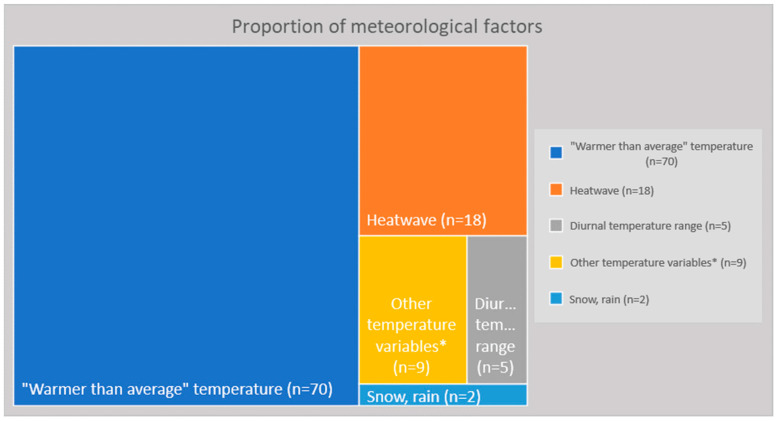
Proportion of meteorological factors used by the authors. ‘Warmer than average’ was usually reported by the authors as high percentiles in relation to a threshold. * Other temperature variables: temperature variability (TV), temperature change between neighboring days (TCN), daily excess hourly heat (DEHH), irregular daily variation.

**Figure 6 ijerph-20-01190-f006:**
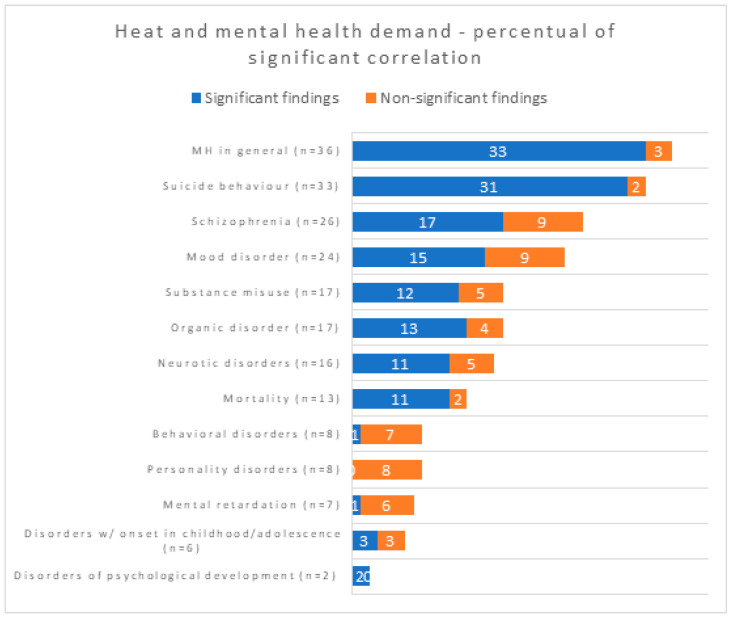
Heat and MH according to the percentage of studies found to be statistically significant (CI 95%).

**Table 1 ijerph-20-01190-t001:** Eligibility criteria: inclusion and exclusion criteria for the selection of papers.

Inclusion Criteria	Exclusion Criteria
Literature and Population	
- Peer-reviewed and original articles. - All countries. - All age groups. - Both sexes. - With pre-existing disorder or with first diagnosis. - Languages: English, Spanish, Portuguese, or German.	- Full text not available. - Non-peer reviewed articles, conference presentations, opinion papers, reviews. - Grey literature (books, government reports, interviews, etc.).
Exposures	
- All variables considered consequences of climate change or climate change-sensitive (heat, heatwaves, extreme events, sea level rising, etc.).	- Studies related to other biological species. - Direct interventions, e.g., psychotherapy or educational measures. - Studies that do not state a comparative value or threshold, in case of hot days OR the used classification for heatwaves.
	- Air pollution or quality of air as isolated exposure were excluded, for being a cause of CC, and not a consequence of it. - Seasonality as the only exposure variable.
Outcomes	
- Admission numbers to urgent care and emergency units or other sources of health systems use (ambulance dispatch, phone calls, etc.) due to mental health symptoms (ICD-10 from F00–F99) or suicide attempt (ICD-10 from X60–X84). - Either the formally diagnosed or simply the symptoms considered, by the time of admission, compatible to such disorders. - Completed suicide numbers. - Mortality in patients with a primary MH disorder that was disrupted, according to the authors, by any CC variable.	- All health outcomes not related to mental health. - Passive measurement of symptoms by questionnaires or subjective perception. - Anxiety, depression, and post-traumatic stress symptoms due to extreme events measured passively with questionnaires or self-perception during community assessments. - Violence or aggression against a third party.

**Table 2 ijerph-20-01190-t002:** General characteristics of all the included studies. * Weather variables included: snow and rain, temperature variability, temperature change between neighboring days, and daily excess hourly heat. The order studies were placed in the table was, as follows: MH in general or multiple types of disorders and, then, the sequence of specific ICD-10th, from F00 to F99. Mortality and suicide were placed next. Lastly, the extreme event-related outcomes.

	Exposure					MH Outcome												MH Service Demand							
	Warmer than average	Heatwave	Other weather variables *	Diurnal Temperature Range (DTR)	Extreme events	MH in general	Organic disorders	Substance misuse	Schizophrenia	Mood disorders	Neurotic disorders	Behavioural disorders	Personality disorders	Mental retardation	Autism spectrum	Behavioural disorders w/onset during childhood/adolescence	Suicide behavior	Emergency department visit	Hospital admission	Ambulance dispatch	Outpatient visits	Suicide	Mortality	Phone calls or services in general	Measure of outcome
No. of studies (n)	70	18	11	5	12	38	17	18	25	27	18	6	5	4	2	4	42	20	34	3	1	34	15	6	
Lee et al. (2018), South Korea [34]	x					x	x		x	x	x							x							Relative risk
Chan et al. (2018), Japan [35]	x					x		x											x						Relative risk
Zhang et al. (2020), China [36]	x						x		x	x	x										x				Odds ratio
Nori-Sarma et al. (2022), USA [37]	x					x		x	x	x	x		x			x	x	x							Incidence rate ratio
Trang et al. (2016), Vietnam [38]		x				x	x	x	x	x	x	x	x	x					x						Relative risk
Trang et al. (2016), Vietnam [39]	x					x				x								x							Relative risk
Basu et al. (2017), USA [40]	x	x				x					x						x	x							% change
Yoo et al. (2021), USA [41]	x					x	x	x	x	x	x							x							Relative risk
Hansen et.al. (2008), Australia [42]	x	x				x	x	x	x	x	x	x	x	x	x	x			x				x		Incidence rate ratio
Middleton et al. (2021), Canada [43]	x		x			x											x							x	Incidence rate ratio
Vida et al. (2012), Canada [44]	x					x												x							Incidence rate ratio
Wang et al. (2013), Canada [45]	x					x		x	x	x	x							x							Relative risk
Niu et al. (2020), China [46]	x					x		x	x	x	x							x							Relative risk
Liu et al. (2018), China [47]		x				x												x							Odds ratio
Sun et al. (2021), USA [48]	x					x												x							Excess relative risk
Shiue et al. (2016), Germany [49]			x				x	x	x	x	x	x												x	Correlation
Settineri et al. (2016), Italy [50]	x					x		x	x	x	x							x							Correlation
Yoo et al. (2021), USA [51]	x					x	x	x	x	x	x							x							Relative risk
Peng et al. (2017), China [52]	x					x													x						Relative risk
Schmeltz et al. (2017), USA [53]	x						x	x	x	x	x	x	x	x		x			x						Risk ratio
Min et al. (2019), China [54]	x					x												x	x						Relative risk
Bundo et al. (2021), Switzerland [55]	x	x				x	x	x	x	x	x	x	x	x	x	x			x						Relative risk
Xu et al. (2018), Australia [56]		x				x												x							Relative risk
Wondmagegn et al. (2021), Australia [57]			x			x												x							Attributable risk
Nitschke et al. (2007), Australia [58]		x				x													x	x			x		Incidence rate ratio
Dang et al. (2022), Vietnam [59]	x	x				x		x	x										x						Relative risk
Carlsen et al. (2019), Sweden [60]	x					x												x							%change
Khalaj et al. (2010), Australia [61]	x					x												x							Relative odds
Williams et al. (2011), Australia [62]	x					x												x	x						Incidence rate ratio
Calkins et al. (2016), USA [63]	x					x		x																x	Relative risk
Wei et al. (2019), USA [64]	x		x				x												x						Razard ratio
Culqui et al. (2017), Spain [65]	x						x												x						Relative risk
Xu et al. (2019), Australia [66]		x					x												x				x		Relative risk
Gong et al. (2022), England [67]	x						x											x							Relative risk
Zhang et al. (2016), Australia [68]		x					x												x						Odds ratio
Pan et al. (2021), China [69]			x	x					x										x						Relative risk
Zhao et al. (2016), China [70]				x					x										x						Relative risk
Yi et al. (2019), China [71]	x								x										x						Relative risk
Pan et al. (2019), China [72]	x								x										x						Attributable and relative risk
Pan et al. (2021), China [73]	x								x										x						Relative risk
Zhao et al. (2016), China [74]			x	x					x										x						Relative risk
Tang et al. (2021), China [75]			x						x										x						Relative risk
Sung et al. (2011), Taiwan [76]	x			x					x										x						Relative risk
Shiloh et al. (2004), Israel [77]	x								x										x						Correlation
Wang et al. (2018), China (135) [78]	x								x										x						Relative risk
Chen et al. (2018), Taiwan [79]	x									x									x						Hazard ratio
Parker et al. (2016), Australia [80]	x									x									x						Correlation
Medici et al. (2016), Denmark [81]	x									x									x						Regression correlation
Volpe et al. (2006), Brazil [82]	x									x									x						Correlation
Shapira et al. (2004), Israel [83]	x									x									x						Correlation
Volpe et al. (2010), Brazil [84]	x		x							x									x						Correlation
Medici et al. (2016), Denmark [85]			x							x									x						Correlation
Zanobetti et al. (2013), USA [86]	x						x																x		Odds ratio
Ho C. H. & Wong M. S. (2019), China [87]	x					x																	x		Incidence risk ratio
Stivanello et al. (2020), Italy [88]	x					x																	x		Odds ratio
Åström et al. (2015), Italy and Sweden [89]		x				x																	x		Relative risk
Kim et al. (2015), South Korea [90]	x					x	x	x		x							x					x	x		Relative risk
Rey et al. (2007), France [91]		x				x																	x		Relative mortality ratio
Page et al. (2012), England [11]	x						x	x															x		Relative risk
Rocklov et al. (2014), Sweden [92]		x				x																	x		Odds ratio
Bark, Nigel (1998), USA [93]		x				x																	x		Relative risk
de’Donato et al. (2007), Italy [94]	x									x													x		%change
Kollanus et al. (2021), Finland [95]		x				x																	x		%change
Gu et al. (2020), China [96]	x					x																	x		Relative risk
Florido Ngu et al. (2021), 60 countries [97]		x															x					x			Incidence risk ratio
Kubo et al. (2021), Japan [98]	x																x			x					Relative risk
Santurtún et al. (2018), Spain [99]	x																x					x			%change
Williams et al. (2015), New Zealand [100]			x														x		x						Increase in incidence
Grjibovski et al. (2013), Kazakhstan [101]	x																x					x			%change
Bär et al. (2022), Switzerland [102]	x																x					x			Relative risk
Chau et al. (2020), China [103]	x																x					x			Bayesian information criterion
Lin et al. (2008), Taiwan [104]	x																x					x			Correlation
	Warmer than average	Heatwave	Other weather variables *	Diurnal Temperature Range (DTR)	Extreme events	MH in general	Organic disorders	Substance misuse	Schizophrenia	Mood disorders	Neurotic disorders	Behavioural disorders	Personality disorders	Mental retardation	Autism spectrum	Behavioural disorders w/onset during childhood/adolescence	Suicide behavior	Emergency department visit	Hospital admission	Ambulance dispatch	Outpatient visits	Suicide	Mortality	Phone calls or services in general	Measure of outcome
Helama et al. (2013), Finland [105]	x																x					x			Correlation
Ruuhela et al. (2009), Finland [106]	x																x					x			Regression coefficient
Aguglia et al. (2021), Italy [107]	x																x					x			Correlation
Barve et al. (2021), India [108]	x																x					x			%change
Akkaya-Kalayci et al. (2017), Turkey [109]	x																x					x			Correlation
Mueller et al. (2011), Germany [110]	x																x					x			%change
Pan et al. (2022), Japan [111]	x																x					x			Relative risk
Burke et al. (2018), USA and Mexico [112]	x																x					x			%change
Preti et al. (2007), Italy [113]			x														x					x			Correlation
Kim et al. (2018), 12 countries [114]	x																x					x			Relative risk
Hu et al. (2020), China [115]		x															x			x					Relative risk
Cheng et al. (2021), USA [116]	x																x					x			Incidence risk ratio
Page et al. (2007), England and Wales [117]	x	x															x					x			%change
Casas et al. (2021), Belgium [118]	x																x					x			Risk ratio
Sim et al. (2020), Japan [119]	x																x					x			Relative risk
Gaxiola-Robles et al. (2013), Mexico [120]	x																x					x			Coefficient of determination
Yarza et al. (2020), Israel [121]	x																x					x			Odds ratio
Likhvar et al. (2010), Japan [122]	x																x					x			Risk estimate
Deisenhammer et al. (2003), Austria [123]	x																x					x			Relative risk
Holopainen et al. (2014), Finland [124]				x													x					x			Correlation
Bozsonyi et al. (2020), Hungary [125]	x																x					x			Stationary R squared
Begum et al. (2022), USA [126]					x	x		x		x	x						x	x	x						Risk Ratio
Phillippi et al. (2019), USA [19]					x	x		x		x	x													x	Descriptive comparison
Wu et al. (2021), China-Lu’na [127]					x				x										x						Relative risk
Reifels et al. (2015), Australia [128]					x					x	x													x	Incidence Rate Ratio
Shih et al. (2020), Taiwan [129]					x					x	x	x					x							x	%increase in incidence
Krug et al. (1998), USA [130]					x												x					x			%change
Matsubayashi et al. (2012), Japan [131]					x												x					x			%increase in incidence
Lee et al. (2019), South Korea [132]					x												x					x			Relative risk
Richardson et al. (2020), India [133]					x												x					x			Relative risk
Alam et al. (2022), India [134]					x												x					x			Relative risk
Hanigan et al. (2012), Australia [135]					x												x					x			Relative risk
Horney et al. (2022), USA [134,136]					x												x					x			%change and rate difference

**Table 3 ijerph-20-01190-t003:** Extreme events.

	Disorder Group	N.	Main Results (* CI 95%)
EDV	MH in general	1	Risk ratio 1.32 * (1.24–1.40)—hurricane [126]
	Substance misuse	1	Risk ratio 1.44 * (1.23–1.65)—hurricane [126]
	Mood disorders	1	Risk ratio 1.59 * (1.39–1.80)—hurricane [126]
	Neurotic disorders	2	Risk ratio 0.79 * (0.72–0.86)—hurricane [126]|Risk ratio 0.80 * (0.73–0.87)—hurricane [126]
	Suicide behavior	1	Risk ratio 1.76 * (1.72–1.80) [126]
HA	MH in general	1	Risk ratio 1.08 * (1.05, 1.10)—hurricane [126]
	Substance misuse	1	Risk ratio 1.12 * (1.04–1.21)—hurricane [126]
	Schizophrenia	1	RR 1.056 * (1.003–1.110)—extreme precipitation [127]
	Mood disorders	1	Risk ratio 1.67 * (1.47–1.87)—hurricane [126]
	Neurotic disorders	2	Risk ratio 1.26 * (1.21–1.31)—hurricane [126]|Risk ratio 0.85 * (0.77–0.94)—hurricane [126]
	Suicide behavior	1	Risk ratio 0.68 * (0.62–0.63)—hurricane [126]
	Substance misuse	1	Increase in number of visits + 66%—flood [19]
Health services in general	Mood disorders	3	IRR 2.57 * (1.60, 4.14)—bushfire and flood [126]|%increase in incidence 308%—hurricane [129]|Increase in number of visits + 44%—flood [19]
	Neurotic disorders	3	IRR 2.06 * (1.21, 3.49)—bushfire and flood [126]|%increase in incidence 307%—hurricane [129]|Increase in number of visits +62%—flood [19]
	Behavioral disorders	1	%Increase in incidence 356%—hurricane [129]
	Suicide behavior	1	%Increase in incidence 18.8%—hurricane [129]
Suicide	Suicide behavior	6	RR 1.15 * (1.08–1.22)—extreme dry weather [135]|RR 4.23 * (1.28–13.93)—drought [134]|%change 18.7% *—extremely wet [133]|RR 1.198 * (1.065–1.347)—Asian dust storm [132]|Increase in incidence 0.6% *—different extreme events [131]|%change 14.8% * (5.4–24.2)—different extreme events [130]

Main results, divided by type of service used and, still, by the MH condition that was demanded. Results presented in red are the significant negative associations. * Statistically significant result (CI 95%). Abbreviations: EDV—emergency department visit, HA—hospital admission, RR—relative risk, IRR—incidence rate ratio.

**Table 4 ijerph-20-01190-t004:** Meteorological factors.

	Disorder Group	N	Main Results (* CI 95%)
EDV	MH in general	3	RR 1.158 * (1.092–1.227) [34]|IRR 1.08 * (1.07–1.09) [37]|%change 4.8% * (3.6–6.0) [40]
	Organic disorders	3	RR 1.045 * (1.029–1.061) [67]|RR 1.31 * (1.22–1.41) [51]
	Substance misuse	4	IRR 1.08 * (1.07–1.10) [37]|RR 1.30 * (1.18–1.42) [51]|RR 1.14 * (1.03–1.32) [45]|RR 3.021 * (1.601–5.703) [46]
	Schizophrenia	3	IRR 1.05 * (1.03–1.07) [37]|RR 2.49 * (1.69–3.69) [45]|RR 1.27 * (1.17–1.38) [51]
	Mood disorders	5	IRR 1.07 * (1.05–1.09) [37]|RR 1.05 * (1.01–1.09) (39)|RR 1.68 * (1.17–2.40) [45]|RR 1.33 * (1.03–1.71) [51]
	Neurotic disorders	5	IRR 1.07 * (1.05–1.09) [37]|%change 5.7% * (3.8–7.6) [40]|RR 1.12 * (1.00–1.27) [45]|correlation r = 0.1947* [50]|RR 1.27 * (1.19–1.36) [51]
	Disorders w/onset during childhood/adolescence	2	IRR 1.11 * (1.05–1.18) [37]
	Suicide behaviour	2	IRR 1.06 * (1.01–1.12) [37]|%change 5.8% * (4.5–7.1) [40]
HA	MH in general	4	RR 1.09 * (1.03–1.15) [35]|RR 1.266 * (1.074–1.493) [52]|RR 1.04 * (1.02–1.07) [55]|RR 1.62 * (1.36–1.93) [59]
	Organic disorders	6	RR 1.84 * (1.66–2.03) [53]|HR 1.12 * (1.09–1.15) [64]|RR: 1.30 * (1.12–1.52) [65]|RR 1.51 * (1.02–1.126) [66]|OR 8.33 * [68]|IRR 1.213 * (1.091–1.349) [42]
	Substance misuse	3	RR 1.13 * (1.00, 1.27) [35]|RR 1.38 * (1.30–1.47) [53]|RR 2.21 * (1.55–3.15) [59]
	Schizophrenia	15	RR 1.10 * (1.04–1.15) [55]|RR 1.50 * (1.20–1.86) [59]|RR 1.195 * (1.041–1.372)# [69]|RR 1.021 * (1.007–1.035)% [69]|RR 1.319 * (1.129–1.540)$ [70]|RR 1.187 * (1.057–1.332)# [70]|RR 1.07 * (1.04–1.11) [78]|RR 1.373 * (1.168–1.614) [71]|RR 1.005* (1.003–1.008) [72]|RR 1.38 * (1.13–1.69) [73]|RR 1.159 * (1.050–1.279)# [74]|RR 1.111 * (1.002–1.231)$ [75]|RR 1.189* (1.051–1.279)! [75]|RR 1.08 * (1.04–1.11) [76]|Correlation r 0.35 * [77]
	Mood disorders	4	RR 1.34 * (1.05 1.71) [35]|IRR 1.034 * (1.009–1.05) [62]|Regression coefficient 0.12 * [81]|Pearson’s correlation r: 0.27 * [83]|IRR 1.091 * (1.004–1.185) [42]
	Neurotic disorders	1	IRR 1.097 * (1.018–1.181) [42]
	Disorders of psychological development	1	IRR 1.641 * (1.086–2.480) [42]
	Disorders with onset during childhood/adolescence	1	RR 1.29 * (1.09–1.54) [55]|IRR 0.578 * (0.349–0.955) [42]
	Suicide behavior	1	%change 0.7% * (0.003–0.011) [100]
Mortality	MH in general	9	IRR 1.033 * (1.004–1.062) [87]|OR 1.055 * (1.024–1.086) [88]|RR 1.33 * (1.10–1.61) [89]|RR 1.04 * (1.01–1.07) [90]|RMR 1.23 * [91]|RR 1.049 * (1.02–1.078) [11]]|OR 1.099 * (1.027, 1.175) [92]|RR 1.38* [93]|%change 29.7% * (21.3–38.6) [95]
	Organic disorders	3	Alzheimer’s disease OR 1.08 * (1.04–1.12) [86]|Dementia OR 1.04 * (1.02–1.09) [86]|RR 1.03 * (1.00–1.07) [11]|RR 269% * (76–665%) [66]
	Substance misuse	2	RR 1.07 * (1.02–1.13) [90]|RR 1.20 * (1.08 1.35) [11]
	Mood disorders	2	OR 1.083 * [88]|%change 166% * (35–424) [94]
Suicide	Suicide behavior	24	RR 0.97 * (0.95–0.99) [90]|IRR 1.035 * [97]|IRR 0.932 * [97]|%change 2.1% * (0.4–3.8%) [101]|RR 1.34 * (1.19–1.52) [102]|association 0.2012 * [104]; Pearson’s correlation 0.617 * [105]|regression coefficient 0.829 * [106]|correlation 0.75 * [107]; Spearman correlation 0.213 * [109]|%change 0.9% * [110]|RR 1.35 * (1.26–1.45) [111]|%change 0.68% * (0.53%–0.83%) [112]|%change 2.1% * (1.2–3.0%) [112]|correlation coefficient 0.55 * [113]|RR 1.33* (1.30–1.36) [114]|IRR 1.0082 * (1.0025–1.0140) [116]|%change 46.9% (15.6–86.8) [117]|risk ratio 2.16 * (1.28–3.63) [118]|RR 1.26 * (1.22–1.29) [119]|coefficient of determination R^2^ 0.64 * [120]|OR 1.59 * (1.22–2.08) [121]|risk estimate 0.0435 * [122]|RR 1.12 * (1.02–1.24) [123]|Pearson’s correlation 0.428 * [124]
Ambulance dispatch	Suicide attempt	2	RR: 1.11* (1.07–1.15) [98]|RR 4.53* (1.23–16.68) [115]

Main results, divided by type of service used and, still, by the MH condition that was demanded. * Statistically significant result (CI 95%). Abbreviations: EDV—emergency department visit, HA—hospital admission, RR—relative risk, IRR—incidence risk ratio, OR—odds ratio, RMR—relative mortality ratio.

## Data Availability

Not applicable.

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
