# Peer review of "A Systematic Literature Review of the Impact of Climate Change on the Global Demand for Psychiatric Services"

_ijerph, 2023, doi:10.3390/ijerph20021190_

Round 1

Reviewer 1 Report

The manuscript “A systematic literature review of the impact of climate change on the global demand for psychiatric services” addresses an important emerging topic on the diverse impacts of climate change on health. However, the methods used are not current within the field of environmental health systematic reviews, and the lack of a full study protocol limits the ability to assess this manuscript’s conclusions.

There is no systematic review protocol accessible in PROSPERO with the registration number provided (CRD42022353023) nor were any of the climate and mental health reviews available in PROSPERO by the authors of this manuscript. I am also concerned that per the naming convention of PROSPERO this number was not assigned until 2022, and given the time required to conduct these reviews I question if the protocol was registered prior to initiating the review. Revisions to a protocol are allowed (and even encouraged) during the execution of a systematic review, provided that is documented along the way (vs. only posting the final protocol once the review is complete). Without seeing the full protocol and any amendment notes, I cannot determine if this was done.

The quality assessment tool used is a score-based tool which are no longer the type most accepted in the environmental health field (see Risk of Bias methods such as those developed by the National Toxicology Program https://ntp.niehs.nih.gov/whatwestudy/assessments/noncancer/riskbias/index.html). The methods section says it was used by two reviewers, but without a tailored protocol it is uncertain if they both rated studies independently and how they came to a consensus. Often these tools need to be tailored for a specific topic to ensure consistency in these subjective ratings. This rating was only used as an exclusion factor prior to data extraction which is an unusual approach and per Figure 2 it only excluded 1 study. Why was this quality assessment not included or used when considering the consistency of results across studies? It was not mentioned again after the screening step, and this is not the preferred approach to assessing study quality in systematic reviews (https://prhe.ucsf.edu/navigation-guide and https://systematicreviewsjournal.biomedcentral.com/articles/10.1186/s13643-020-01490-8).

The exclusion of air pollution studies is poorly justified – there is a large body of literature on criteria air pollutants such as ozone in combination with heat and humidity and it seems as if any study including air pollution was excluded from this review.

The Results section thoroughly described the findings but lacks synthesis nor did it identify how future studies could be strengthened to address the limitations in the current literature. The Discussion also delves into topics beyond the scope of this review, such as the carbon footprint of health care and mechanisms of heat impacts on mental health.

Minor Concerns:

Some of the English is awkwardly worded and could be improved (e.g., line 45 “effects of CC on psychiatry reach not only punctually individuals”).

Why did figure 1 only go through 2014 other than to compare a low point on the graph in figure 1b? Also the colors in 1a make it hard to distinguish between Respiratory and Other (and if the Other line is around 20 in 2014, this implies that the majority of Other is Mental Health). Why not just put mental health from figure 1b on the same graph as 1a and continue both out through 2022? It is also unclear how this figure supports the preceding statement about individual vs community level impacts of CC on mental health.

The introduction could be condensed to focus on the primary topics that this systematic review is capable of addressing – the introduction is broad and raises many issues beyond the scope of this review.

In the Methods section, on line 120 it states the terms were combined with Boolean operators but does not specify which ones (AND/OR/NOT).

How was the order of studies determined for Table 2 – it appears random and some sort of clustering or sorting by publication date might make it more interpretable. The lumped Other Weather Variable column mixes some temperature related-exposures that seem more relevant to the other columns.

Tables 3 and 4 data would be better displayed as a forest plot graph that would more clearly show the direction, size, and confidence limits of the estimated effect across the studies.

Minor typo: Line 64 “varying up to on in five people” should be one not on

Author Response

Dear reviewer 1, we appreciate very much and thank the acknowledgment of the topic’s relevance, as well as the comments and constructive criticism.  We hope we had now solved the major flaws pointed out in the previous manuscript.

  1. “The manuscript “A systematic literature review of the impact of climate change on the global demand for psychiatric services” addresses an important emerging topic on the diverse impacts of climate change on health. However, the methods used are not current within the field of environmental health systematic reviews, and the lack of a full study protocol limits the ability to assess this manuscript’s conclusions”.

R. Dear reviewer, we appreciate your concern with the important steps taken during the methodology approach. We carefully followed the PRISMA protocol (Preferred Reporting Items for Systematic Reviews and Meta-Analyses) which is among the most used and followed guidelines. In order to illustrate it, we took the liberty to cite an example of 2 papers, Thompson et. al (2018) and Charlson et. al. (2021), where this same guideline was used. PRISMA provide us with a clear methodology, as it follows: two independent readers must exclude papers firstly for i) title and abstract. In this step, if included by any of the two reviewers, the paper is kept for further analysis. Next phase, ii) full text selection, the two readers, independently, go through the papers and, in case of disagreement, the third reader is required. Once studies were searched for full text, according to PRISMA, the chosen quality assessment tool should be used specifically on these studies (also by the two reviewers independently, with a third one being required if necessary). Finally, we tried to edit the text, in order to make it reproductible and clear for the reader (‘Materials and methods’ – section 2).

References:

-PRISMA:  Page, M. J., Moher, D., Bossuyt, P. M., Boutron, I., Hoffmann, T. C., Mulrow, C. D., ... & McKenzie, J. E. (2021). PRISMA 2020 explanation and elaboration: updated guidance and exemplars for reporting systematic reviews. bmj372.

-Thompson, R., Hornigold, R., Page, L., & Waite, T. (2018). Associations between high ambient temperatures and heat waves with mental health outcomes: a systematic review. Public health161, 171-191.

-Charlson, F., Ali, S., Benmarhnia, T., Pearl, M., Massazza, A., Augustinavicius, J., & Scott, J. G. (2021). Climate change and mental health: a scoping review. International journal of environmental research and public health18(9), 4486.

  1. “There is no systematic review protocol accessible in PROSPERO with the registration number provided (CRD42022353023) nor were any of the climate and mental health reviews available in PROSPERO by the authors of this manuscript. I am also concerned that per the naming convention of PROSPERO this number was not assigned until 2022, and given the time required to conduct these reviews I question if the protocol was registered prior to initiating the review. Revisions to a protocol are allowed (and even encouraged) during the execution of a systematic review, provided that is documented along the way (vs. only posting the final protocol once the review is complete). Without seeing the full protocol and any amendment notes, I cannot determine if this was done”.

R. We thank the reviewer 1 for checking the PROSPERO. Our registration at their website dates on August 22nd, 2022. We took the liberty to upload the registration PDF. Concerning how fast the review was carried out, we should have clarified the reasons and we apologize for that: the topic this review approaches - CC and MH - is exactly the same from my (Julia Feriato Corvetto) doctoral thesis, which I am currently working on. Therefore, we already had most of the background literature stored and systematized. Once we had the idea of carrying this SLR by narrowing down the topic to 'health services use', in July, we quickly organized the keywords and distributed the work each author would develop. In August, when we were ready to begin, we registered the manuscript on PROSPERO. About the need for updates on PROSPERO website, we sincerely apologize, because we were not yet aware it was possible and recommended to do so. Thank you for this information and on future systematic reviews we will surely update it.

  1. The quality assessment tool used is a score-based tool which are no longer the type most accepted in the environmental health field (see Risk of Bias methods such as those developed by the National Toxicology Program https://ntp.niehs.nih.gov/whatwestudy/assessments/noncancer/riskbias/index.html). The methods section says it was used by two reviewers, but without a tailored protocol it is uncertain if they both rated studies independently and how they came to a consensus. Often these tools need to be tailored for a specific topic to ensure consistency in these subjective ratings. This rating was only used as an exclusion factor prior to data extraction which is an unusual approach and per Figure 2 it only excluded 1 study. Why was this quality assessment not included or used when considering the consistency of results across studies? It was not mentioned again after the screening step, and this is not the preferred approach to assessing study quality in systematic reviews (https://prhe.ucsf.edu/navigation-guide and https://systematicreviewsjournal.biomedcentral.com/articles/10.1186/s13643-020-01490-8)”.

R. Thank you for the concern with this important step of the SLR. We acknowledge your comment and we argue that, according to PRISMA, only once studies are full text reviewed for the inclusion and exclusion criteria, we then need to use the quality assessment tool. In our case, we used the tool in all the 106 included studies to check for quality of design, methodology and results. It would be an extensive and non-justifiable work if we had to check the quality of the 7,432 screened studies. Furthermore, NIH developed and personalized the quality assessment tools according to the design of the analyzed study. We used either the ‘quality assessment tool for Observational Cohort and Cross-Sectional Studies’ or ‘for Case-Control studies’, in order to reduce bias. We again take the liberty to cite Thompson et. al. (2018) and Charlson et. al. (2021), as they used the NIH tool in their paper as well. Also, we added to the text (first paragraph of Results section) the number of graded 'poor', 'fair' and 'good' studies. Finally, we were not aware of the possible bias of using this current tool and, therefore, we acknowledged this information on the ‘Strengths and Limitations’ section (lines 936-939).

References:

-Thompson, R., Hornigold, R., Page, L., & Waite, T. (2018). Associations between high ambient temperatures and heat waves with mental health outcomes: a systematic review. Public health161, 171-191.

-Charlson, F., Ali, S., Benmarhnia, T., Pearl, M., Massazza, A., Augustinavicius, J., & Scott, J. G. (2021). Climate change and mental health: a scoping review. International journal of environmental research and public health18(9), 4486.

  1. The exclusion of air pollution studies is poorly justified – there is a large body of literature on criteria air pollutants such as ozone in combination with heat and humidity and it seems as if any study including air pollution was excluded from this review”.

R. We appreciate the comment. We argue though that air pollution would be a cause of climate change, and not a consequence of it, as the other variables: heat, extreme events, etc. A more detailed explanation was added to the Table 1, so the reader can understand the reasons as well.

  1. The ‘Results’ section thoroughly described the findings but lacks synthesis nor did it identify how future studies could be strengthened to address the limitations in the current literature. The Discussion also delves into topics beyond the scope of this review, such as the carbon footprint of health care and mechanisms of heat impacts on mental health”.

R. The knowledge gaps were added and emphasized on the text. Additionally, based on your suggestions, we aimed to synthesize the ‘results’ at the beginning of each section (‘extreme events’ or ‘meteorological factors’). Also, the mechanisms of heat impact on mental health were substantially reduced. We opted to keep at least part of it, since, in our perspective, delineating mechanisms may be a helpful way of showing policymakers and health professionals where they should act, in order to reduce acute consultations. As an example, medical doctors can, in some circumstances, adjust certain psychotropic types or doses, in order to prevent heat-related illnesses and, therefore, consultations. Finally, mentioning the carbon footprint of healthcare facilities may call attention, in our opinion, for the self-enforcing cycle of this intense hospital demand.

  1. “Some of the English is awkwardly worded and could be improved (e.g., line 45 “effects of CC on psychiatry reach not only punctually individuals”)”.

R. Thank you for the correction. We opted to delete the sentence from the text, adapting the paragraph in a more objective format. English language was reviewed by the authors, as well.

  1. “Why did figure 1 only go through 2014 other than to compare a low point on the graph in figure 1b? Also, the colors in 1a make it hard to distinguish between Respiratory and Other (and if the Other line is around 20 in 2014, this implies that the majority of Other is Mental Health). Why not just put mental health from figure 1b on the same graph as 1a and continue both out through 2022? It is also unclear how this figure supports the preceding statement about individual vs community level impacts of CC on mental health”.

R. Thank you very much for this valuable point of view. We followed it and created another graphic (figure 1) in order to contemplate the concerns pointed out by the reviewer and to translate the information in a more visible way for the reader. The last statement, about individual vs community levels of impact, was already adapted, as well.

  1. “The introduction could be condensed to focus on the primary topics that this systematic review is capable of addressing – the introduction is broad and raises many issues beyond the scope of this review”.

R. We thank the reviewer for this comment. We altered the 'introduction section' in order to narrow down the approached topics and make it more objective for the review’s purposes.

  1. “In the Methods section, on line 120 it states the terms were combined with Boolean operators but does not specify which ones (AND/OR/NOT)”.

R. The specific Boolean operators are listed on the supplementary materials. Thank you for pointing it out. We had forgotten to mention they were described in the supplementary materials, but we now added this information in this revised manuscript (lines 154-155).

  1. “How was the order of studies determined for Table 2 – it appears random and some sort of clustering or sorting by publication date might make it more interpretable. The lumped Other Weather Variable column mixes some temperature related-exposures that seem more relevant to the other columns”.

R. Dear reviewer, our intention was to join the similar disorders, so the reader could compare how they were approached, and for that we followed the ICD-10th revision. On the top of the table, we placed the ‘mental disorders in general’ – the ones not specified by the respective authors – and the studies comprising various MH disorders. Next, we selected the disorders listed on ICD-10th as F00-F09, and subsequently, until F90-F99. Mortality, suicide and extreme events are the last ones. In order to facilitate the interpretation and comprehension, we carefully clarified it on subtitle of Table 2. Also, the “Other weather variables” were placed together because they are very specific types of measurement, which would possibly disguise the reader from the already better scientifically delineated factors (“hot temperature”, “heatwave”, “diurnal temperature range” and “extreme events”). Besides, since delineating each of the measurements was not our focus, we opted to keep the main factors more visible.

  1. Tables 3 and 4 data would be better displayed as a forest plot graph that would more clearly show the direction, size, and confidence limits of the estimated effect across the studies”.

R. We thank the reviewer for this valuable idea. We followed it and created the new figure 5, that can be found on the new manuscript. We opted to keep the tables 3 and 4, because we sincerely believe it might be interesting for the reader to see i) the results from different measurements of outcome (RR, risk ratio, %change, etc.), that cannot be represented in a forest plot. Also, ii) being able to delineate how the health services were demanded according to each type of disease might be clarifying.

  1. Minor typo: Line 64 “varying up to on in five people” should be one not on”

R. Thank you for pointing out this mistake. We reviewed and corrected it in the text. We opted to change this sentence to a clearer version, using percentage instead.

Reviewer 2 Report

The authors have chosen a pertinent area for review. I have a few concerns with the methods and presentation of findings. 

1. It is unclear how climate change variables have been operationalized. So, when the authors talk about 'hot temperatures' increasing mental health problems, what does 'hot temperatures' mean is not stated. Similarly, for other climate change variables. 

2. It is a little confusing if the authors are looking at health services utilization, or the increase in prevalence of disorders. They raise the importance and differences in studying these in the introduction, but the same has not been brought out in the results/ discussion. The results and discussion seem to have collapsed these two aspects. 

3. In the introduction, authors have also raised the issue of high versus low/middle income countries and the challenges in measuring health service utlisation in them. Even though they found some studies from low/middle income countries, the differences have not been brought out in the results.

4. Too many types of 'disorders' (some diagnostic categories, some specific behavioral phenotypes) have been used as outcome variables. I feel this becomes very confusing for the reader. The authors could use identify presentations of specific interest and focus the review around them. 

5. The manuscript has a lot of data, lot of significant findings but lacks in clarity, interpretative synthesis and drawing specific conclusions for the objectives of the study. Authors need to pay more attention to collating the data in a scientific manner. 

Author Response

Dear reviewer 2, thank you very much for your review. Your comments were constructive, kindly guiding us to the effective improvement of the manuscript. We hope we were able to address carefully all your concerns and advises.

  1. “It is unclear how climate change variables have been operationalized. So, when the authors talk about 'hot temperatures' increasing mental health problems, what does 'hot temperatures' mean is not stated. Similarly, for other climate change variables”. 

R. Thank you for pointing out how this information was still unclear. We addressed this issue in clear statements in the text (introduction and methodology sections – lines 44-45, 164-169 – and table 1), in order to define it better. The CC variables were considered to be all the possible components that can be potentially influenced or worsened by CC: meteorological factors and extreme events. In this context, air pollution was not included in this review, since it is considered to be one of the causes of CC, instead of a consequence of it. Still, ‘hot temperature’, ‘heat’ or ‘warmer than average’ were temperature values above a certain threshold specified by the author, as it is defined by the MeSH term ‘hot temperature’: “Presence of warmth or heat or a temperature notably higher than an accustomed norm” - https://www.ncbi.nlm.nih.gov/mesh/?term=hot+temperature.

  1. “It is a little confusing if the authors are looking at health services utilization, or the increase in prevalence of disorders. They raise the importance and differences in studying these in the introduction, but the same has not been brought out in the results/ discussion. The results and discussion seem to have collapsed these two aspects”. 

R. Thank you for pointing it out. We aimed to look specifically at services utilization. Based on your comment, we tried to delineate better these two distinct aspects on the results and discussion.

  1. “In the introduction, authors have also raised the issue of high versus low/middle income countries and the challenges in measuring health service utlisation in them. Even though they found some studies from low/middle income countries, the differences have not been brought out in the results”.

R. Thank you for this important perception. A paragraph addressing this previously missing aspect was added to the text. Please, see lines 744-750. Additionally, we pointed this lack of study as a gap of knowledge (862-867 and 897-907).

  1. “Too many types of 'disorders' (some diagnostic categories, some specific behavioral phenotypes) have been used as outcome variables. I feel this becomes very confusing for the reader. The authors could use identify presentations of specific interest and focus the review around them”. 

R. Dear reviewer, we acknowledge the great number of “disorders” that were listed in the free text and we also understand that it might become confusing for the reader. On the other hand, thought, we sincerely believe that by focusing only on the “most common” ones – and removing the others –, we would prevent the reader from receiving this information. These disorders are already less approached and poorly addressed by science and we aimed to do the opposite and call attention for them. Still, in order to make it easier and clearer in the text, we reorganized the order in which they were placed: the subgroups in the results section are now placed in order of relevance. Lastly, we added two summarizing paragraphs at the beginning of each subgroup “extreme events” and “meteorological factors”. The phenotype psychosis was also removed from the table 2, in order to follow only the ICD-10th disorders.

  1. “The manuscript has a lot of data, lot of significant findings but lacks in clarity, interpretative synthesis and drawing specific conclusions for the objectives of the study. Authors need to pay more attention to collating the data in a scientific manner”. 

R. Thank you for this very pertinent comment. In order to improve the display of the data in a more objective manner, we created a forest plot (figure 5) in addition to tables 3 and 4. Also, we aimed to carefully synthesize the main results in the first paragraphs of the 'discussion section', so the relevant information can be comprised and reminded according to the objectives of this review (ex.: Lines 712-750).

Reviewer 3 Report

Dear authors,

Congratulations for your work. In addition to being well structured and written, the manuscript addresses a relevant and current topic. It follows an adequate methodology for a systematic review, which is well described allowing the replication of the study.

I have just a few suggestions/comments:

-          Join the paragraphs of lines 81-87, 518-528 and 529-538;

-          Verify the formatting of the first paragraph of point 3.2 (line 352);

-          Line 364: I believe it should be (n=33) and not [33].

-          Line 447: they say "eleven out of sixteen", but they refer to having 14 studies on neurotic and anxiety disorders.

Kind regards,

Reviewer

Author Response

Dear reviewer 3, we kindly appreciate and thank very much the acknowledgment of the relevance of our review. We have carefully read all of your comments and suggestions and, hopefully, we addressed all of them.

  1. “Congratulations for your work. In addition to being well structured and written, the manuscript addresses a relevant and current topic. It follows an adequate methodology for a systematic review, which is well described allowing the replication of the study. Join the paragraphs of lines 81-87, 518-528 and 529-538”.

R. Thank you. We agreed with the suggestions. We joined the paragraphs 81-87 and 529-238. Also, we joined and shortened the paragraphs 518-528 in accordance.

  1. “Verify the formatting of the first paragraph of point 3.2 (line 352)”.

R. Thank you for the correction. We had not noticed that and now we addressed it correctly.

  1. “Line 364: I believe it should be (n=33) and not [33]”.

R. Thank you for calling our attention to this point. We, again, had not noticed that and now we edited it properly.

  1. “Line 447: they say "eleven out of sixteen", but they refer to having 14 studies on neurotic and anxiety disorders”.

R. We apologize for this mistake and we appreciate the careful reading of our manuscript. We corrected the text in accordance to your comment.

Round 2

Reviewer 2 Report

Thank you for submitting a revised version of the manuscript. 

However, it was a little challenging for me to note the changes in the manuscript because the line numbers mentioned in the response to comments did not match the line numbers in the submitted revised version. 

I am still not convinced with the clarity in presentation. For example, operationalizing of climate change variables. When you say 'exposure' to hot temperatures, what exactly was measured? What has been classified by the authors/ the reference studies as hot temperatures? Similarly for other CC variables. 

The authors have just listed the climate change variables and then listed the prevalence parameters for MH conditions. However, how these two were associated is unclear. When you have reported a relative risk for a MH condition, in terms of a CC variable, what were the numerator/ denominator in calculation. There is a world-wide increase in MH problems. So, the authors will have to clarify much more how they are linking these increases to the impact of CC. 

The topic chosen for review is highly relevant and timely, but I am not convinced the review approaches its objectives well enough, for the reasons stated above. 

Author Response

Dear reviewer 2,

we would like to sincerely thank you for the opportunity to clarify and, hopefully, strengthen the methodology used in this present review, mostly on the operationalization of variables. Please, find the point-by-point responses below: 

1. “Thank you for submitting a revised version of the manuscript. 

However, it was a little challenging for me to note the changes in the manuscript because the line numbers mentioned in the response to comments did not match the line numbers in the submitted revised version.” 

R. Dear reviewer 2, we apologize for the difficulties found when searching the mentioned lines and we thank you for looking for them through the text. In case any of these referenced lines were impossible to be found, we took the liberty to list the edited elements, that are related to the operationalization of CC variables. Please, find it below:

Lines 44-45

"This wide range of Climate Change-sensitive variables – weather components potentially worsened by Climate Change (CC) – imposes consequences on human health."

Lines 164-169

"The CC variables were considered to be all the possible components that can be potentially influenced or worsened by CC: meteorological factors and extreme events [2]. In this context, air pollution was not included in this review, since it is considered to be one of the causes of CC, instead of a consequence of it. Still, ‘hot temperature’, ‘heat’ or ‘warmer than average’ were temperature values above a certain threshold specified by the author."

Table 1

Inclusion criteria (exposures): "All variables considered consequences of climate change or climate change-sensitive (heat, heatwaves, extreme events, sea level rising, etc.)."

2. “I am still not convinced with the clarity in presentation. For example, operationalizing of climate change variables. When you say 'exposure' to hot temperatures, what exactly was measured? What has been classified by the authors/ the reference studies as hot temperatures? Similarly for other CC variables. 

The authors have just listed the climate change variables and then listed the prevalence parameters for MH conditions. However, how these two were associated is unclear. When you have reported a relative risk for a MH condition, in terms of a CC variable, what were the numerator/ denominator in calculation.”

R. Dear reviewer, we sincerely appreciate your concern with the operationalization of variables, as this is an important step of the methodology. We tried to be very objective and careful when carrying out the ‘Data and Methods’ section and, therefore, we had previously listed all the assessed information from the studies, including the mentioned CC variables. This new detailed table was now included on the ‘Supplementary materials’ of the revised manuscript and we apologize for not considering adding it earlier. The table is extremely long and carries a lot of data, so we mistakenly thought it would only confound the reader. Please, find it on the ‘Supplementary materials’ as table F. We are also uploading the Excel Table, for a better visualization. 

The table presents the different definitions of ‘hot temperature’, ‘heatwave’, ‘extreme events’ and other meteorological factors used by the authors.

These CC variables were included in the review considering the latest report of the ‘Intergovernmental Panel on Climate Change’, which counts on a specific chapter: “Weather and Climate Extreme Events in a Changing Climate” (IPCC, 2021). The initially retrieved articles were then filtered to capture only papers approaching variables susceptible to CC – considered as a direct consequence of it.

To exemplify two reviews that used similar variables, we take the liberty to cite Thompson et. al. (2018) and Cianconi et. al. (2020). The first ones carried out a SLR on the impact of Heat and Heatwave over MH and the latter, a Systematic Descriptive Review on the MH impacts of CC in general.

We cannot affirm that every single extreme weather event or episode of heat approached by the 105 studies were directly caused by CC. What we do know, based on strong scientific evidence, is that these events are becoming more frequent and more intense (IPCC, 2021) and that they have the potential to increase health systems demand worldwide – more than it would increase in the absence of CC.

References:

I. Seneviratne, S.I., X. Zhang, M. Adnan, W. Badi, C. Dereczynski, A. Di Luca, S. Ghosh, I. Iskandar, J. Kossin, S. Lewis, F.  Otto, I.  Pinto, M. Satoh, S.M. Vicente-Serrano, M. Wehner, and B. Zhou, 2021: Weather and Climate Extreme Events in a Changing Climate. In Climate Change 2021: The Physical Science Basis. Contribution of Working Group I to the Sixth Assessment Report of the Intergovernmental Panel on Climate Change [Masson-Delmotte, V., P. Zhai, A. Pirani, S.L. Connors, C. Péan, S. Berger, N. Caud, Y. Chen, L. Goldfarb, M.I. Gomis, M. Huang, K. Leitzell, E. Lonnoy, J.B.R.  Matthews, T.K. Maycock, T. Waterfield, O. Yelekçi, R. Yu, and B. Zhou (eds.)]. Cambridge University Press, Cambridge, United Kingdom and New York, NY, USA, pp. 1513–1766, doi:10.1017/9781009157896.013.

II. Thompson, R., Hornigold, R., Page, L., & Waite, T. (2018). Associations between high ambient temperatures and heat waves with mental health outcomes: a systematic review. Public health161, 171-191.

III. Cianconi, P., Betrò, S., & Janiri, L. (2020). The impact of climate change on mental health: a systematic descriptive review. Frontiers in psychiatry11, 74.

3. “There is a world-wide increase in MH problems. So, the authors will have to clarify much more how they are linking these increases to the impact of CC. 

The topic chosen for review is highly relevant and timely, but I am not convinced the review approaches its objectives well enough, for the reasons stated above.” 

R. We firstly thank the reviewer 2 for pointing out the relevance of the topic, as well as for indicating how MH problems are rising worldwide. We kindly argue though that the increase in MH disorders worldwide is uniform and a long-term trend. The heat – or other CC related-event –, contrarily, is considered a short-term exposure. With different types of methodologies – e.g., regression analysis with Distributed Lag Non-Linear Model – the authors relied on tools to control for these long-term confounders and, consequently, focused only on the exposure of interest (e.g., heat). We also argue that, in order to make sure the authors did control for confounders, we used NIH Quality Assessment Tool – specifically for cross sectional or for case-control study – that includes the following questions:

i) “Were key potential confounding variables measured and adjusted statistically for their impact on the relationship between exposure(s) and outcome(s)?” - Quality Assessment Tool for Observational Cohort and Cross-Sectional Studies.

ii) “Were key potential confounding variables measured and adjusted statistically in the analyses? If matching was used, did the investigators account for matching during study analysis?” - Quality Assessment Tool for Case-Control Studies.

Finally, the increase in admission numbers presented by the studies was usually compared to a normal baseline – number of patients with MH disorders that looked for healthcare in an average day (in the absence of CC-related events). If this baseline is higher today than it was 20 years ago for example, the increase measured by the studies might have been potentialized, since more people with MH problems were exposed to those events. But we still kindly argue that it does not change the fact that health services worldwide will be more intensively demanded. Contrarily, in our perspective, this high prevalence of MH diseases only contributes to strengthen the reasons in order to perform this SLR, as we mention in the text:

Lines 79-85 (in track-change modus)

“This burden may be still intensified, due to the high rates of psychiatric disorders worldwide. Since 2005, there was a growth of approximately 15% on the prevalence of common mental diseases and, among youth, they occupy the leading cause of health affection in the United States [14,15]. The incidence during lifetime can reach levels up to 50% of the population, depending on the country [16]. Finally, once it is diagnosed, the condition imposes important disability for the patient [17].”

Reference:

I. National Institute of Health (n.d.). Study Quality Assessment Tool. Retrieved December 28th, 2022, from https://www.nhlbi.gov/health-topics/study-quality-assessment-tools